# Increased Cell Proliferation as a Key Event in Chemical Carcinogenesis: Application in an Integrated Approach for the Testing and Assessment of Non-Genotoxic Carcinogenesis

**DOI:** 10.3390/ijms241713246

**Published:** 2023-08-26

**Authors:** Christian Strupp, Marco Corvaro, Samuel M. Cohen, J. Christopher Corton, Kumiko Ogawa, Lysiane Richert, Miriam N. Jacobs

**Affiliations:** 1Gowan Crop Protection Ltd., Harpenden AL5 2JQ, UK; 2Corteva Agriscience, 00100 Rome, Italy; 3Department of Pathology and Microbiology and Buffett Cancer Center, University of Nebraska Medical Center, Omaha, NE 68198, USA; 4Center for Computational Toxicology and Exposure, United States Environmental Protection Agency (US EPA), Research Triangle Park, NC 27711, USA; corton.chris@epa.gov; 5Division of Pathology, National Institute of Health Sciences, Kawasaki 210-9501, Japan; 6Zylan, 67210 Obernai, France; 7United Kingdom Health Security Agency (UK HSA), Radiation, Chemicals and Environmental Hazards, Harwell Innovation Campus, Dicot OX11 0RQ, UK

**Keywords:** non-genotoxic carcinogens, carcinogenicity, mitogenicity, regenerative proliferation, hazard assessment, molecular targets, new approach methods (NAMs)

## Abstract

In contrast to genotoxic carcinogens, there are currently no internationally agreed upon regulatory tools for identifying non-genotoxic carcinogens of human relevance. The rodent cancer bioassay is only used in certain regulatory sectors and is criticized for its limited predictive power for human cancer risk. Cancer is due to genetic errors occurring in single cells. The risk of cancer is higher when there is an increase in the number of errors per replication (genotoxic agents) or in the number of replications (cell proliferation-inducing agents). The default regulatory approach for genotoxic agents whereby no threshold is set is reasonably conservative. However, non-genotoxic carcinogens cannot be regulated in the same way since increased cell proliferation has a clear threshold. An integrated approach for the testing and assessment (IATA) of non-genotoxic carcinogens is under development at the OECD, considering learnings from the regulatory assessment of data-rich substances such as agrochemicals. The aim is to achieve an endorsed IATA that predicts human cancer better than the rodent cancer bioassay, using methodologies that equally or better protect human health and are superior from the view of animal welfare/efficiency. This paper describes the technical opportunities available to assess cell proliferation as the central gateway of an IATA for non-genotoxic carcinogenicity.

## 1. Introduction

Increased cell proliferation has been accepted as a key hallmark and characteristic of cancer for several decades although the concept of cell proliferation as a specific mode of action for carcinogenesis was not well received until the mid-1900s. The relationship between cell proliferation and carcinogenesis was first postulated in a seminal paper published by Armitage and Doll in 1954, which described the increased incidence of cancer with age [1]. In this publication, a major parameter was the rate of cell replication. The authors considered this to be constant throughout life, an assumption that later turned out to be incorrect. The next major contribution was a publication by Knudson in 1971 demonstrating that the incidence of retinoblastoma in both the inherited and sporadic forms of the disease could be explained by a combination of (1) the inheritance or development of one abnormal allele and (2) the development of inactivating mutations in the other allele, secondary to normal cell proliferation and without an environmental influence [2]. This led to the concept of “tumor suppressor genes”, in this instance, the retinoblastoma gene.

The model of interaction between cell proliferation and cell damage was described by Moolgavkar and Knudson utilizing epidemiology data related to breast cancer in women [3], and by Cohen and colleagues based on various chemical carcinogenesis models for rat bladder cancer [4,5,6]. These models showed the role played by increased cell proliferation in previously unexplained carcinogenesis factors such as latency, spontaneous tumors, and the distinction between so-called tumor initiators (genotoxic chemicals) and tumor promoters (chemicals acting by increasing cell proliferation) [6]. Two further major publications by Tomasetti and colleagues in 2015 [7] and 2017 [8] illustrated once again how the interaction between spontaneous cell proliferation and the effects of the environment on cell proliferation (i.e., actual number of replications; see further details below) explains the incidence of tumors in various tissues. 

The concept of cell proliferation in carcinogenesis is founded on several basic principles [4,9,10]. It is well accepted that cancer is due to errors arising stochastically during DNA replication that are not repaired, and that more than one error is necessary for cancer to develop. The errors must occur in single cells, given that cancer is a clonal disease. Furthermore, every time that DNA replicates, although the process is extremely precise, errors occur because of the wide variety of endogenous DNA adducts that are present in cells [11]. There are therefore two ways for an environmental agent to enhance the risk of cancer: (1) by increasing the number of irreversible errors occurring during DNA replication via direct DNA damage (genotoxic agents), and/or (2) by increasing the number of DNA replications and thereby the opportunity for spontaneous errors to occur (cell proliferation-inducing agents). 

In this second scenario, enhanced cell proliferation is caused by increased cell births or decreased cell deaths. Increased cell births are the result of direct mitogenicity (e.g., estrogen stimulation of breast cells) or compensatory regeneration following increased cell deaths. Increased cell deaths can be due to necrosis (e.g., chloroform and liver and kidney tumors) or increased apoptosis (e.g., fumonisin and kidney tumors in rats) [12,13]. Decreased cell deaths are produced by reducing either differentiation or decreasing apoptosis. This leads to an accumulation of cells, and (as the total number is higher at the start of the next cell replication cycle) providing more opportunities for spontaneous errors even if the rate of cell proliferation is the same as that of controls. There are specific examples of decreased cell death possibly contributing to increases in cell number, for example, some evidence in animal models exposed to nuclear receptor activators [14,15,16,17]. 

Another consideration is the understanding of the potential causal role of oxidative stress in non-genotoxic carcinogenesis, which is not completely understood and remains controversial. There is often an association between pre-carcinogenic lesions and oxidative stress [18], often in association with inflammation. However, direct oxidants (e.g., hydrogen peroxide, H_2_O_2_) or redox cyclers (such as acetaminophen, due to its metabolite, or paraquat) are not carcinogenic. Hence, further research is needed to determine if oxidative stress is a causal mechanism, or a consequence of cytotoxicity (cell death) and a biomarker for critical inflammatory processes that may be associated with regenerative proliferation (for example, via mitogenic cytokines).

The key parameter of cell proliferation in non-genotoxic carcinogenesis is the absolute number of replications rather than the rate [19]. Enhanced rate is frequently associated with an increase in the number of cells that are present (hyperplasia), but the two do not necessarily occur together. An example is liver carcinogenesis in rodents caused by constitutive androstane receptor (CAR) [14,15] or peroxisome proliferator-activated receptor alpha (PPARα) [16,17] activators. In both cases, there is an initial increase in the rate of cell proliferation; then, the rate returns to control levels within 1–4 weeks of continued exposure. However, in that time, there is an accumulation of cells, and even if they continue to proliferate at the rate of control hepatocytes, this represents a significant increase in the overall number of DNA replications. 

Numerous direct modes of action may lead to higher cell proliferation, but the specifics vary from one case to the next. In animal models used for cancer hazard assessment of environmental chemicals and pharmaceuticals, the focus should be on the key events that lead to increased cell proliferation and the subsequent formation of tumors. This understanding allows the relevance for humans to be evaluated and forms the basis of the International Program on Chemical Safety (IPCS) mode of action/human relevance framework [13,20,21,22,23]. 

In addition to the fact that genotoxicity and/or increased cell proliferation are the bases for carcinogenesis, there can be significant synergy between these modes of action. This was illustrated in the so-called “mega mouse” study evaluating the effects of 2-acetylaminofluorene in mouse liver and bladder carcinogenesis [24], an instance where a single chemical can have multiple modes of action, including genotoxicity and toxicity at higher doses, with consequent increased cell proliferation. Cases are also known where different agents produce the two effects. For example, when the genotoxic compound N-(4-(5-Nitro-2-furyl)-2-thiazolyl)formamide(FANFT) and non-genotoxic sodium saccharin are administered together, there is a significant increase in bladder tumors of the rat [25]. In contrast, when rodents are exposed to either low-dose FANFT or sodium saccharin alone in the diet, no detectable incidences of tumors are produced. Cigarette smoke contains a complex mixture of chemicals that can produce genotoxicity and increased cell proliferation [26,27]. 

The assessment of cell proliferation in vivo is being used more frequently in modern agrochemical and pharmaceutical early development. Knowledge of cell proliferation responses helps to prioritize compounds for development and can contribute to the information necessary to establish a toxicity threshold in relation to non-genotoxic carcinogenicity hazard assessment. Cell proliferation status has also been applied in regulatory waiver applications for rodent cancer bioassays (i.e., the “Rethinking Carcinogenicity Assessment for Agrochemicals Project” (ReCAAP) [28]).

In 2020, an expert group of the Organization for Economic Cooperation and Development (OECD) developing an integrated approach for the testing and assessment (IATA) for non-genotoxic carcinogens (NGTxC) published a consensus paper [29] describing the overarching IATA, with the molecular initiating events (MIE) of cellular metabolism and receptor interactions, followed by the early key events of inflammation, immune dysfunction, mitotic signaling, or cell injury, leading to (sustained) proliferation, and then morphological transformation leading to tumor formation (Figure 1).

Overall, increased cell proliferation is pivotal as a key event in the assessment of non-genotoxic chemical carcinogenesis, either as a direct biological target (i.e., MIE) or as a consequence of other MIEs and downstream activations. 

The present paper describes the methods available to assess cell proliferation as a central gateway in non-genotoxic human carcinogenicity. The objective is to translate and apply this knowledge to the IATA for non-genotoxic carcinogenicity and thereby increase regulatory confidence [29]. The final goal is to integrate the cell proliferation key event into an OECD-endorsed IATA. This would increase the ability to predict human non-genotoxic carcinogenicity better than with the rodent cancer bioassay, with methodologies that equally or better protect human health, and are superior from the view of animal welfare/efficiency. The present paper uses the agreed OECD definition of new approach methods (NAMs), whereby any mechanistic and refined approach to assess the human carcinogenic potential other than traditional rodent cancer bioassays—e.g., grouping and read-across, defined approaches, in vitro test guidelines (TG), in silico models, and animal tests if they serve to reduce or refine other animal tests—are considered to be NAMs [31].

## 2. Assessing Cell Proliferation In Vivo and In Vitro 

### 2.1. General Aspects

Cell proliferation was historically assessed by counting mitotic figures, but this technique is no longer used since it is laborious and not as reproducible as newer immunohistochemical methods. With immunohistochemistry, proliferation can be visualized and quantified by either (a) nucleoside analog incorporation into newly synthesized DNA (tritiated thymidine, or 5-bromo-2’-deoxyuridine (BrdU)), or (b) cell cycle-associated protein expression (nuclear antigens: Ki-67 and proliferating cell nuclear antigen (PCNA)) [32]. 

#### 2.1.1. Nucleoside Analog Incorporation into Newly Synthesized DNA

During genome replication (the S-phase of the cell cycle), DNA polymerases incorporate nucleosides into new strands of DNA. The nucleosides made available to the cell can be radiolabeled, or nucleoside analogs can be used. Both lead to detectable signals for cells with newly synthesized DNA (cells that have undergone division). 

In vivo, nucleosides can be administered by intraperitoneal (i.p.) or intravenous (i.v.) application or by means of subcutaneous (s.c.) pumps, to interfere as little as possible with the application of the test chemicals [32,33]. In vitro, excess nucleoside analogs are added to cell cultures and allowed to incubate for multiple days, after which the excess is washed away [34,35,36]. 

In the BrdU incorporation assay, antibodies are used (against the nucleoside analog taken up into freshly synthesized DNA), and the readout can be visual (immunohistochemistry on tissue slices) or by enzyme-linked immunosorbent assay in vitro.

In the tritiated thymidine (^3^H-Thy) assay, incorporated radioactivity is quantified by either using autoradiography (in vivo) or a liquid scintillation counter (in vitro). This method quantifies overall division compared to a control group and is commonly regarded as reliable and accurate. Drawbacks are that radioactive reagents must be handled and disposed of with caution, and no additional assays can be performed with or after ^3^H-Thy incorporation (it is an endpoint assay), even in vitro, because the assay extracts DNA from other cellular components that are then washed away in the process. The tritiated ^3^H-Thy assay is, therefore, less utilized to date.

#### 2.1.2. Cell Cycle-Associated Protein Expression and Corresponding mRNA

PCNA synthesis begins in the late G1 phase, peaks during the S phase, declines in G2, and is virtually absent in the M phase. Therefore, PCNA is often used as a marker for cell proliferation due to its vital role in DNA replication. A major advantage of using PCNA antibodies is that they can be applied to fixed tissue without detrimental effects on histological architecture and does not require administration of an exogenous marker such as BrdU in vivo. The major limitations of the PCNA technique are varied staining intensities of nuclei, the dependence of staining on the fixation method (methanol fixation results in the labeling of cells in the S phase, whereas other types of fixations permit the staining of all cycling cells), and PCNA’s longer half-life (20 h, it can be expressed even when the cells reach G0) [32].

Ki-67 antigen is found in all proliferating cells and is expressed in all phases of the cell cycle (G1, S, and G2) and mitosis. Therefore, antibodies against the Ki-67 protein are widely used. The percentage of cells staining positive for the Ki-67 antigen is called the Ki-67 labeling index, directly related to cell proliferation rate. Ki-67 staining protocols vary based on the detection method: flow cytometry or colorimetric approaches, or immunohistochemical methods in tissues. In the immunohistochemical methods, the cells are fixed and become permeabilized, and then anti-Ki-67 antibody is added. A secondary antibody recognizing the Ki-67 antibody is added, after which quantitative analysis is performed. Immunostaining of Ki-67 antigen with monoclonal antibody can detect Ki-67 equivalents in several different species such as cattle, dog, horse, sheep, rodents, and humans, and can be used with formalin-fixed, paraffin-embedded tissue. 

In recent years, several reports have mentioned that the expression of Ki-67 mRNA could be used as a relevant (additional) biomarker for Ki-67 protein expression and cell proliferation [37,38]. Major advantages are increased sensitivity compared to immunohistochemistry and the possibility to determine the expression of other genes in the same total RNA samples obtained from cultures of specific cell types, allowing for the calculation of correlation coefficients between genes of control and treated samples. For tissue analysis, it is important to identify the proliferative response in the specific target cell type of the tumorigen, for example, hepatocytes in the liver for hepatocellular tumors. 

#### 2.1.3. The Tools of Choice

Currently, BrdU and Ki-67 are the most frequently used methods for the detection and quantification of cell proliferation in vivo and in vitro. The most obvious difference between the two is when to measure: Ki-67 provides a snapshot of proliferative activity shortly before endpoint determination, while BrdU incorporation can capture a cumulative incorporation over the time of administration [39,40]. 

Increased proliferation numbers over background can either be sustained, for example, because of constant cell damage and repair, or mitogenic and have a peak while the organism is adapting to the xenobiotic load. The latter is commonly observed in rodents upon activation of nuclear receptors. When the time of peak effect is known or the peak time of proliferation is to be established, Ki-67 may be the tool of choice, as serial time points can be assessed, and administration of nucleoside analogs to the animals or cell cultures is not needed. Another advantage of Ki-67 is that it does not require the administration of an agent to monitor proliferation; thus, the staining can be used on archival tissues for retrospective evaluation, with the assumption that a peak of proliferation can be seen. However, it requires several groups of animals to be sacrificed at multiple time points per dose level. If this is not needed, BrdU has the advantage of detecting the cumulative number of cell divisions over a period, so missing a peak of proliferation is less likely. With this approach, fewer animals are used since the cumulative proliferation effect can be established in a single group of animals per each dose level. An additional advantage includes confidence in the assessment of proliferation peaks that may be dependent on the kinetics (target tissue concentration) of the xenobiotic inducer. However, an additional substance (BrdU) is introduced into the test system. Also, in vivo administration is associated with additional workload (jugular vein or s.c. continued pump infusion or serial i.p. administration every few hours), represents a potential source of uncertainty if i.p. administration is used (this can be overcome by assessing a constantly proliferating tissue, such as small intestine, to verify successful administration), and introduces potential technical pitfalls (BrdU may clog s.c. minipumps) as well as potential interferences (narcosis needed for the implantation of minipumps). Hence, BrdU may be more suitable if the research question is whether there is increased proliferation rate within a time interval of interest. If the peak of proliferation rate is already established and the research question is the timing and quantity of the proliferative response, Ki-67 may be the most suitable biomarker. 

#### 2.1.4. Target Organs and Target Cell Types: Some Case Examples

The responses obtained with both methods correlate well between in vivo and in vitro [32]. While it is in principle possible to investigate proliferation in vivo in any tissue at any time by immunohistochemical staining of tissue slices or biopsy material, the prerequisites for the proliferation need to be understood before moving into the specific assessment of cell proliferation in vitro. It is important to establish the target cell type (e.g., Club cell) and not just an organ (e.g., lung) in general. Standard histopathology is typically adequate but may require immunohistochemistry or electron microscopy [32].

Examples are discussed below to illustrate, based on mode of action, which aspects are critical to address. An important aspect is timing. While induction of metabolic enzymes can be associated with a strong proliferative peak (in particular, prior to xenobiotic load adaptation and flattening off thereafter), constant damage/repair or cell signaling lead to later and more sustained proliferation.

Relatively early in the development of cancer bioassays, in the 1940s, it was noted that a proliferative stimulus plays an important role in carcinogenesis. At that time, it was observed that proliferation can shorten the time to detectable skin tumors when a DNA mutagen was provided. This was investigated with the help of solvents that either directly induced proliferation or damaged skin to activate repair proliferation; however, the most specific experiments were those where proliferation was induced by local wounding and resulted healing [41].

In the gastrointestinal tract, for example, as well as urinary bladder and liver, chemical-induced cytotoxic damage is typically more sustained. Here, the test chemical (or a metabolite thereof) can either reach a cytotoxic concentration (e.g., in the duodenum by direct contact upon oral exposure) [42,43,44] or create physical damage due to crystal formation (e.g., in urine) [45]. Furthermore, the metabolite may not be generated in the target tissue, but elsewhere (e.g., liver), and transported to the target site (e.g., urine). Proliferation occurs in response to constant and persistent damage and is more readily detected than a proliferative peak. Also, while in vitro cell proliferation systems may be possible, today, they are not sufficiently optimized with respect to application to non-genotoxic carcinogenicity (see Section 2.3). Furthermore, the retention of metabolic competence or incorporation of knowledge of metabolite generation and excretion is mandatory. For example, when considering the bladder, the concentration ability in the kidney is needed. Also, for calculus-related chemicals, the chemical or metabolite may not even be involved. 

In the mammalian liver, numerous chemicals activate nuclear receptors such as the CAR, pregnane X receptor (PXR) [14,15,46,47], or PPARα [16,17,48]. Activation triggers a proliferative stimulus in rodents to the hepatocyte, leading to the production of more hepatocytes. The typical toxicological picture is liver weight increase, centrilobular/periportal hypertrophy in shorter-term studies, and formation of liver hyperplasia and ultimately hepatocellular adenomas and carcinomas when exposure is extended to lifetime. An increase in proliferation rate can be detected in the form of a peak within the first 1–14 days while the animal adapts to the xenobiotic load. The time point may vary depending on the specific marker, chemical, cell type, test species, and gender. Proliferation has also been measured in primary hepatocytes in vitro. These cells, ideally grown on an extracellular matrix, are known to retain some metabolic capacity, and may be used as a test system for investigating the responses of different species to chemicals for their proliferative potential [36,49]. Both in vivo and in vitro, while the rate peaks, the number of hepatocytes increases and therefore the number of replications continues to be higher than controls. The number of replications is the key parameter.

Another example is the induction of phase II metabolism in rats (in particular uridine 5’-diphospho-glucuronosyl transferase, UDP-GT), leading to increased clearance of thyroid hormones [50]. The resulting induction of thyroid-stimulating hormone (TSH) puts a continuous grow stimulus on the thyroid, inducing colloidal changes, proliferation, and finally tumors. The proliferation is at a low level, but constant; this is not easy to detect, as the signal is close to the background. However, the integration of liver weight/histopathology/UDP-GT induction, in combination with increased TSH and thyroid weight/histopathology, allows a clearer assessment of the situation.

As another example, in the mouse lung, several chemicals trigger the induction of Cyp2f2 cytochrome P450 in Club cells, and these chemicals may act as true mitogens or are activated to reactive or cytotoxic metabolites. These cytotoxic agents damage the Club cells, and the response is a strong proliferative stimulus. The typical toxicological picture for cytotoxic chemicals is macrophage invasion in short-term studies and lung hyperplasia/adenoma for mitogenic or cytotoxic chemicals in mice only (not rats) if exposure is extended for lifetime. Proliferation can be detected as a peak in short-term exposure studies while the animals adapt to xenobiotic exposure. No in vitro models detecting this proliferation have been developed to date, but the mode of action (MoA) is well understood, and an in vitro model’s development is possible since the target cell and the mechanism are known [51,52,53].

### 2.2. Challenges of the In Vivo Evaluation of Cell Proliferation

For in vivo studies, it is necessary to consider the appropriateness of the evaluation method for each target organ or target cell to be analyzed. It is critical that the evaluator be blinded to the group that the samples come from, and standardized methods for the selection of fields to be evaluated should be stated in the protocol. Examples of causes that may induce false positives and/or false negatives are discussed below.

In case the proliferation rate of the target cells is fast, it may be possible to reduce the number of cells that need to be counted. For example, in the normal colon of rats injected i.p. with BrdU 1 or 2 h before sacrifice, approximately 4.1–6.7/pit [54] or 12.8–15.8% [55] were positive, respectively, when they were counted in 12 or 10 crypts. The number of cells, as a denominator of the evaluation, differs between organs/cells with rapid cell proliferation (e.g., intestine) and those with slow cell proliferation (e.g., liver and bladder) [32]. For example, the ratio of BrdU-positive cells in the normal liver and bladder of rats injected i.p. with BrdU 1 h before sacrifice is about 0.28–0.48% [54] and 0.06–0.32% [54,56], respectively. In such cases, for sufficient statistical power, at least 1000–3000 cells should be evaluated for comparison. The impact of the error in the number of positive cells depending on the selection of view may become small if the number of positive cells is high. However, when the ratio of positive cells is low, fewer counts will result in a greater error due to the difference depending on the field of view (Figure 2). Also, to avoid selection bias, it is recommended that the slides be coded so that the individual counting the cells is blinded to the group the sample is from. The evaluation fields should be as similar as possible with respect to histological location.It is desirable to count only target cells and not multiple cell types. For example, for the evaluation of proliferation in pancreatic endocrine cells, only islet cells should be assessed. Exocrine tumors can arise from ductular or acinar cells. Although the phenomenon is rarely seen in rats, the human pancreas is more likely to develop carcinogenesis from duct-derived cells than from islets or acinar cells. If there are reasonable animal models for human pancreatic cancer, it is appropriate to evaluate only the ductal cells (Figure 3). However, this may be decided on a case-by-case basis. 

Although promising at the current time, automated image analysis systems are not considered sufficiently reliable on their own without quality control by a pathologist.

3.In principle, it is considered more accurate to evaluate proliferation using the number of cells as the denominator, but when cells exist at the same density, it is also possible to examine per unit area. Consideration is nevertheless required when the size or the density of target cells differs amongst groups. For example, if there is hepatocyte hypertrophy, the number of cells per unit area will become smaller, the denominator will be overestimated, and the positive ratio will become smaller than it is (Figure 4).4.If there is a difference in cell proliferation across the organ, it would be necessary to evaluate the fields evenly to reflect the whole organ. For example, the epithelial cells in the gastrointestinal tract and urinary bladder have certain proliferative zones consisting of immature cells located mostly in the basal layer of the urothelium or in the crypts of the intestine, but middle height in the mucosal glands in the stomach. A bias may occur in the ratio between the different layers depending on how the organ is cut, so care must be taken to make the sectioning representative of the normal distribution of target cells (Figure 5). To avoid this bias, a greater number of cells should be evaluated compared to more homogenous tissues.5.When tissues with a mixture of lesions are anticipated, it is important to evaluate cell proliferation in each of the histological lesions such as normal-like areas, hyperplasia, benign tumors, and malignant tumors. If the histopathological lesion is not considered, the evaluation field should reflect the existing lesion as a whole and care must be taken to avoid bias. Hyperplastic areas indicate higher cell proliferation than normal-like areas, usually with an increased rate of proliferation as well as an increased number of cells.6.Evaluation of BrdU (or/and ^3^H-labeling) indices should be compared to a control group administered under the same conditions. Factors such as the time after administration and the time of day (diurnal changes), and exposure concentration may impact results. Basically, if enough BrdU or ^3^H is administered, even if the amount of uptake into each nucleus during the DNA synthesis varies, the mitotic phase changes. The labelling reagents may not be taken up by cells outside the S phase, resulting in inconsistent labeling values. Ki-67 evaluation may also be affected by the duration of the fixation time and the specimen preparation conditions, for example, so the comparison to the concurrent control group is crucial. Samples for the determination of Ki-67 should ideally not be left in the fixative for more than 72 h before trimming and embedding.

It is recommended to control the appropriate administration of nucleoside analogs by sampling of a tissue with a higher basal proliferation rate (e.g., small intestine), as incorrect administrations (i.p., i.v.), or clogging of osmotic minipumps is possible. 

7.In some cases, the cell proliferation rate is transiently induced in the initial period of administration of the test chemical, and the labeling rate increases, after which the proliferation ratio returns to normal levels. However, an increase in cell number results in an increase in the denominator, and the number of proliferating cells is still high even though the labeling rate looks decreased. For example, the meaning may be different between 10% positive normal-looking urothelial cells in three layers and 10% positive urothelial cells in six layers due to simple hyperplasia with increased cell density. In that case, the number of positive cells per unit basement membrane length is more than doubled and may be of different significance (Figure 6).

Additional considerations when assessing cell proliferation include the use of a standardized evaluation strategy that minimizes biases and that can visualize nondividing cells using double staining techniques if necessary. 

If rare tumors are to be evaluated in a 2-year bioassay, large groups (50 animals) are needed to detect significant differences in tumor incidences. Conversely, for cell proliferation evaluations, fewer animals are needed (based on practical experience and for adequate statistical power, this can be typically between 8 to 15 per group, depending on the specific tissue).

### 2.3. Challenges and Opportunities for the In Vitro Evaluation of Cell Proliferation 

Several in vitro assays are available for detecting either direct (gene mutation; DNA or chromosomal damage) or indirect (DNA repair) genotoxicity and have been successfully implemented into testing strategies in the last decades. The same applies to in vitro cytotoxicity assays, which have been successfully developed (cell membrane functionality, cellular metabolic activity, adenosine triphosphate (ATP) content). Currently, a test chemical with the potential of DNA damage or cell death can be detected using the available battery of tests/endpoints. For the less common mechanism, inhibition of apoptosis, several reliable cell-based tests (caspase activation, annexin V) have been developed and are being critically reviewed in a separate manuscript [57].

Several challenges are yet to be met when assessing the ability of a chemical to trigger cell proliferation. First, cell cultures need to be metabolically competent to reflect the true biological situation. Considerable progress in cell culturing has been made, and partially metabolically competent primary hepatocyte (sandwich-) [58] or three-dimensional spheroid cultures [59] are now available. In addition, methods have been developed to retrofit existing bioassays with metabolic competence using the lid-based alginate immobilization of metabolic enzymes (AIME) method, which adds hepatic metabolism to conventional high-throughput screening platforms [60]. Coculturing methods with immunocompetent (Kupffer) cells have also been developed [61,62]. Still, major questions must be addressed before applying cocultures with immunocompetent cells.

Despite the extensive experience showcased in the published literature for in vivo assessment of proliferation using BrdU or Ki-67 [32], formal validation with reference chemicals (known to increase or not cell proliferation in vivo) is desirable to speed up their regulatory acceptance. In due time, once promising in vitro test methods have been sufficiently optimized, assay set-up with primary cells from both rodent and human origin is also considered critical. The former allows for in vivo–in vitro comparison for a given animal species, and the latter elaborates on in vitro species comparison and extrapolation to humans. Alternatively, or additionally, primary cells isolated from animals knocked out for specific genes can be used to further investigate or confirm essentiality of a pivotal key event in a postulated MoA. 

Furthermore, the effect of the test chemical should be evaluated using wide concentration ranges, including concentrations relevant to the in vivo situation, i.e., at concentrations of the chemical that the target tissue is exposed to at chronic steady state over a range of external doses. If the concentration is high enough, a high proportion of agents will produce cytotoxicity in vitro, which is considered of added value to verify that a reduced or lack of response at lower concentrations is not due to inappropriate testing. The use of metabolically competent test systems may facilitate part of this assessment, as is common practice for genotoxicity or other cytotoxicity animal and human cell-based assays. Access to the relevant exposure, i.e., the concentrations of chemical or metabolite reaching the target tissue, may be complex and may need to be extrapolated from “absorption-distribution-metabolism-excretion” (ADME) studies in rodents (for example, using ^14^C-labeled material), and/or from integrated in vivo toxicokinetic data in other matrices (i.e., in blood, urine, and, if possible, target tissue), and/or using generic or compound-specific physiologically based kinetic (PBK) models, which may include data and models from human test systems [63]. 

In addition to these challenges, specific difficulties are related to the in vitro assessment in cell cultures of response to growth factor- and hormonal factor-induced signals, as well as of regenerative proliferation-associated signals resulting from cytotoxicity. Immortalized, actively proliferating cells may not be an appropriate model system. Primary cells are preferred unless it has been demonstrated that the effect can be reproduced in cell lines. First, as for many other endpoints, mechanistically associated with cell proliferation, such as CAR-, PXR-induced phase I and II enzyme induction in the liver, in vitro responses are either lower than in vivo responses, or need long-term exposure (between 7 and 14 days of daily treatment), or both. Therefore, it is difficult to assess increases in cell numbers in vitro. For this reason, assays for cell proliferation include those detecting DNA synthesis (nucleoside analog incorporation (^3^H-Thy and BrdU), cell cycle-associated mRNA/protein expression (Ki-67, PCNA), and cytoplasmic proliferation-related dyes (carboxy fluorescein succinimidyl ester (CFSE)). Even these assays show some limitations due to their borderline sensitivity. Assays should normally be repeated several times independently using varying numbers of cells to generate a proliferation curve. For example, while the responses to mitogens (e.g., epidermal growth factor (EGF)) are strong and reproducible in cell cultures, this is not the case for regenerative as well as nuclear receptor activation-associated proliferation. It is thus highly recommended to assess the concentration-dependent cytotoxicity profile in parallel to potential regenerative-induced cell proliferation, and for agents activating nuclear receptors, to assess the concentration-dependent specific enzyme inductions in parallel with cell proliferation assays. Furthermore, a set of positive controls needs to be first agreed upon, where possible, with regulators. The sole cell proliferation signal(s) would not allow for any conclusion as to their origin.

Whilst there are several in vitro methods for proliferation, these do require further optimization and reassessment prior to taking forward for TG validation purposes. The process can be facilitated by translational application from in vivo studies.

### 2.4. “Readiness” and Appropriateness Evaluation of Cell Proliferation Assays

A subgroup of the OECD expert group developing the IATA for non-genotoxic carcinogens (NGTxC) critically assessed the readiness and relevance of available assay approaches for the assessment of cell proliferation according to the criteria described in [29]. Table 1 provides an overview of the assessment (conducted by the expert group) of critical aspects and limitations, according to the criteria described above. 

In vivo studies addressing proliferation were ranked by the expert group in category A as addressing the endpoint of interest in fully metabolically and immunocompetent test system including toxicokinetic aspects (i.e., if the agent reaches the target in sufficient concentration). The systems have been described over many decades by different laboratories with consistent responses on prototypical inducers (Table 2) and hence appear ready for formal validation. In vitro assays addressing de novo cell proliferation were ranked in category B/C, as they were found to be more limited in terms of metabolic competence, more challenging in the interpretation of toxicokinetic aspects, and void of immunocompetence. Proliferation needs to be addressed in line with MoAs for tumor formation. In that sense, regenerative proliferation due to sustained cytotoxicity, for example, is not the same as proliferation of cells due to favorable culture conditions. Furthermore, although sometimes marketed as such, cell viability is not the same as cell proliferation, as it also includes cell death. Thus, in vitro assay kits that measure the number of viable cells in culture based upon quantification of the ATP present, as an indicator of metabolically active cells, are not a substitute for measuring cell proliferation and rather assess cytotoxicity. Results have been presented over a few decades by different laboratories, with generally consistent responses, however the weakness due to the absence of metabolic competence and immunological effects makes results less easy to translate to in vivo; hence, conceptual optimization work will be needed before entering formal validation. In vitro assays addressing only cell numbers were assigned to category C and not further evaluated, as they are not sufficiently biologically relevant. An additional need is an agreed list of positive controls. EGF is a single polypeptide of 53 amino acid residues involved in the regulation of cell proliferation and is the classical positive control. It should increase cell proliferation approximately three- to sixfold in mouse and human primary hepatocytes, demonstrating the similar responsiveness of mouse and human primary hepatocyte cultures to a proliferative stimulus.

Table 2 provides some first examples of prototypical chemical inducers and non-inducers of cell proliferation in mammals. Rodent cell proliferation is often species-specific. 

## 3. The Value of Adding “-Omics” to In Vitro/In Vivo Cell Proliferation Assays

To address some of the issues identified above, particularly with respect to elucidating the modulation of genes and pathways associated with cell proliferation, a variety of robust “-omics” approaches, including transcriptomics, proteomics, metabolomics, and epigenomics, are available for uncovering mechanisms underlying carcinogenic responses, including the involvement of cell proliferation. Whilst epigenomics [98] and transcriptomic approaches for cell signaling [99] for the NGTxC IATA are reviewed elsewhere, it is pertinent to this discussion to elaborate on the transcriptomic tools for cell proliferation. Transcriptomics (defined as measuring changes in global RNA levels) is arguably the most mature and widely used of the -omics approaches; transcriptomics has provided a robust body of evidence for chemical and genetic factors responsible for perturbing molecular pathways leading to tumors [98,99,100,101]. There are numerous public (e.g., Gene Expression Omnibus) and private (e.g., Illumina’s BaseSpace Correlation Engine) repositories of transcriptomic data that continue to grow in scale and number, providing opportunities to build, test, and incorporate transcriptomics-based molecular tools into cancer hazard assessment strategies [102]. Evolving computational methods are available to identify differentially expressed genes that can be overlaid onto molecular networks of disease or biological pathways at the systems biology level [103] to formulate hypotheses linking exposure to pathology [104]. However, there is a great deal of variability in dataset interpretation, and for regulatory needs, harmonized approaches in reporting -omics data need to be considered (for which OECD guidance is under development: https://www.oecd.org/chemicalsafety/testing/draft-oecd-omics-reporting-framework-guidance-doc-may-2022.pdf (accessed on 17 August 2023)).

Assessments of global gene expression in hazard studies have provided a wealth of information about the pathways perturbed after chemical exposure. The pathways comprising gene lists include those in public (e.g., Gene Ontology, Molecular Signatures Database) and private (e.g., Ingenuity Pathways Analysis) databases that are used to generate hypotheses as to how chemical exposure could lead to perturbation of molecular targets, pathways of metabolism, and phenotypic effects. Usually, these hypotheses must be followed up with additional research to confirm or refute the predictions, because little is known about the ability of the gene lists to be able to accurately predict an effect using a computational method in a defined context of use. 

There are now opportunities to move beyond these hypothesis-generating tools to be able to more accurately predict the effects of chemical exposure using gene expression biomarkers including those that are linked to the induction of cell proliferation. Gene expression biomarkers (also referred to as “signatures”) consist of sets of genes known or predicted to be regulated by a particular factor or cellular process [105]. These biomarkers are being increasingly recognized by broad sectors of the scientific community to make accurate predictions of key events in adverse outcome pathways underlying chemical carcinogenesis. NAMs applied in in vivo and in vitro contexts that use batteries of biomarkers could be used in IATA strategies, such as those proposed by Oku and colleagues [99], or could have utility in the context of a longer-term outlook [106]. 

A cell cycle proliferation (CCP) gene set was developed that has the potential to predict the cell proliferation status in a number of exposure contexts. The CCP gene set was originally used to assess cell proliferation in human tumors [107]. Genes were initially selected from the Gene Expression Omnibus database and their performance was tested with RNA extracted from 96 commercially available formalin-fixed paraffin-embedded (FFPE) prostate tumor sections. The authors found that the expressions of the genes were highly correlated with each other. The final signature consisted of 31 CCP genes. The study provided strong evidence that the CCP signature score based on the expression of the genes is a robust prognostic marker for cell proliferation and could be used to determine the appropriate treatment for patients with prostate cancer. Subsequently, the CCP gene set was used to assess cell proliferation in another set of prostate cancers [108] and human breast cancers [109]. The CCP gene set has been shown to predict changes in cell proliferation in rodent liver and human cell lines [57], and thus can contribute to the provision of in vivo–in vitro translational evidence for a transition from in vivo to in vitro tools, in the longer term.

A biomarker that could be potentially used to predict cell proliferation in the rat liver was partially characterized and built from transcript profiles from chemical treatments known to induce cell proliferation through different mechanisms including those induced by acetaminophen, carbamazepine, gemfibrozil, phenobarbital, thioacetamide, and Wyeth (WY)-14643 [110]. The gene lists were filtered for those genes found in lists in Molecular Signatures Database (MSigDB) gene sets labeled “cell cycle” (http://software.broadinstitute.org/gsea/msigdb/collections.jsp (accessed on 17 August 2023)). The relationships between the liver-to-body weight changes (LW/BW) and activation of cell proliferation was indirectly assessed using the cell proliferation biomarker. Analysis of 4- to 29-day exposures in the rat liver was performed during a time when the liver increased in size likely due, in part, to increases in cell proliferation. As the LW/BW increased, so did the correlation between the genes altered by chemical exposure, and the cell proliferation biomarker. Ultimately though, the predictive accuracy of the cell proliferation biomarker could not be calculated due to the lack of relevant data to compare the genomic effects with (e.g., immunostaining for PCNA or Ki-67), currently not found in the TG-GATES database. While there are examples of biomarkers accurately predicting molecular effects in mice and rats in vivo, there does not appear to be examples of well-characterized biomarkers with known accuracies that predict cell proliferation in different exposure scenarios. 

Regulatory acceptance of biomarker use, to date, is rare. The toxicogenomic (TGx)-DNA damage-inducing (DDI) biomarker is currently under regulatory review by the United States Food and Drug Administration (US FDA) through the Center for Drug Evaluation and Research Biomarker Qualification Program [111]. Thus far, only the GARD^TM^skin/GARD potency biomarker signature used in conjunction with artificial intelligence to identify skin sensitizers in a human myeloid dendritic-like cell line have been accepted for regulatory studies (OECD TG 442E). However, further promising biomarker test methods on the OECD TG program, intended for guideline adoption, are forthcoming, pending completion of successful validation and peer review (e.g., the ToxTracker assay, a mammalian stem cell-based genotoxicity assay employing six green fluorescent protein reporters specific for DNA damage, oxidative stress, and protein misfolding [112]).

A biomarker predictive of modulation of cell proliferation in vitro would be useful to link chemical exposure, MIE activation, and downstream activation of cell proliferation. Nair et al. [113] identified genes associated with proliferation across more than 40 breast cancer cell lines and built a biomarker of breast cancer proliferation from the cell line transcript profiles. The biomarker accurately estimated proliferation values (*p*-value < 1.79^−5^) using a standard cross-validation procedure. The genes of the proliferation predictor were enriched in cell differentiation, promoter transcriptional regulation, and tissue development. The authors then applied the cell proliferation biomarker to estimate the proliferation levels of more than 1000 breast cancer tumors from the Cancer Genome Atlas (TCGA). The authors found that cell proliferation predicted using the biomarker increased with the tumor’s aggressiveness, as qualified by its stage, grade, and subtype. Future studies might focus on the ability of the cell proliferation biomarker to predict modulation of cell proliferation from high-throughput transcriptomics (HTTr) data as part of a coordinated assessment of molecular targets and phenotypes that could be used to identify chemicals with carcinogenic potential. It is important to recognize, however, that data from cancer cell lines reflect characteristics of the cancer cell and have reduced relevance to the effects in normal cells.

In summary, -omics technologies provide an opportunity to add to carcinogenicity assessment both in a role as a prescreening tool and as supportive evidence. This is further elaborated in Oku et al. [99]. Guidance is under development for consistent reporting of -omics results. 

## 4. Application of Cell Proliferation Assay Tools in the NGTxC IATA: Proposed Way Forward

In the previous sections, we discussed how and what to best measure with respect to in vivo and in vitro cell proliferation assays for the regulatory assessment of NGTxC, extrapolating mainly from the perspective of data-rich substances. We have examined how to generate lines of evidence on cell proliferation (as NGTxC IATA key event) from regulatory toxicity assays (e.g., standard 28- and 90-day tests) and bespoke hypothesis-driven short-term in vivo and in vitro proliferation assays, with consideration to species, dose and/or concentration setting, and kinetics. Strengths and potential shortcomings are identified, together with approaches as to how to experimentally overcome these. In this way, we can progress the development of the cell proliferation interface between in vivo and in vitro scenarios for non-genotoxic carcinogenic hazard assessment. 

Current regulatory use of this evidence is based on established frameworks for Mode of Action/Human relevance assessment (World Health Organization (WHO)/International Program on Chemical Safety (IPCS)), weight of evidence (OECD, European Food Safety Authority (EFSA), and European Chemicals Agency (ECHA)), biological relevance (OECD and EFSA), and uncertainty analysis (e.g., OECD and EFSA). Considering the advanced status of the characterization of these assays and their biological relevance to non-genotoxic carcinogenicity mechanisms, the integration in the upcoming framework designed by the OECD NGTxC IATA project is desirable. A proposed integration is illustrated in Figure 7. The proposed IATA is based on generic key events, which can be addressed with different “assay blocks”, i.e., assays meant to provide evidence on chemical-specific effects on each generic key event. These pieces of evidence can then be assembled to understand potential for the adverse outcome (i.e., carcinogenesis) to occur. Figure 7 shows the potential for the assays described within this paper to be integrated into the original OECD NGTxC IATA described in [29]). These assays provide information on the modulation of the generic key event “(sustained) proliferation” (originally identified as “assay block” 2) and they are subdivided into primary mitogenicity or proliferative response to tissue damage (green central section of Figure 7). Evidence from cell proliferation-related “assay block” 2 could provide evidence that can be integrated with other evidence from other assay blocks within the OECD NGTxC IATA.

(QSAR: quantitative structure activity relationship; CYP: cytochrome P450; GIT: gastrointestinal tract; TG: test guideline).

## 5. Conclusions

Learnings from data-rich substances, such as agrochemicals and pharmaceuticals, with respect to the assessment of cell proliferation can be translated to other substances for which repeated dose testing has been conducted, but the rodent cancer bioassay is not available. It is pertinent to include approaches for prioritization purposes and/or triggering and waiving of in vivo studies under specific strong evidential circumstances, as adopted recently by the International Council for Harmonization of Technical Requirements for Pharmaceuticals for Human Use (ICH) S1B Addendum and Waivers for Testing Pharmaceuticals for Carcinogenicity [114], and retrospective ReCAAP [28]. 

The inclusion of (sustained) cell proliferation as a key event in an IATA for NGTxC was recommended by the OECD expert group. An assessment of the experimental tools currently available to generate relevant evidence has been delineated in this review. Herein, we have provided the essential components that could be utilized directly to augment in vivo 28- and 90-day guideline toxicity studies, providing critical evidence for non-genotoxic carcinogenicity at earlier time points (using fewer animals and shorter times than traditional rodent cancer bioassays). Furthermore, short-term in vivo and in vitro proliferation assays are currently in use in regulatory settings as supplementary evidence contributing to the mechanistic weight of evidence/human relevance assessment of the key event of (sustained) cell proliferation. Albeit missing formal validation, the in vivo assays are more reliable, based on readiness assessment currently ongoing at the OECD level. Primary cell-based in vitro assays would benefit from the development of acceptance criteria for a given culture. Whilst they are currently used by the agrochemical industry to prioritize compounds for development, these tools can also find greater use by regulatory authorities for the assessment of non-genotoxic carcinogenicity to prioritize chemicals of concern. In vitro cell proliferation assays based on immortalized cell lines require substantial optimization and development translated from the in vivo situation. This will provide a strong basis to ultimately enable the transition from in vivo studies to credible in vitro assays in future regulatory settings within the context of the OECD NGTxC IATA framework.

## Figures and Tables

**Figure 1 ijms-24-13246-f001:**
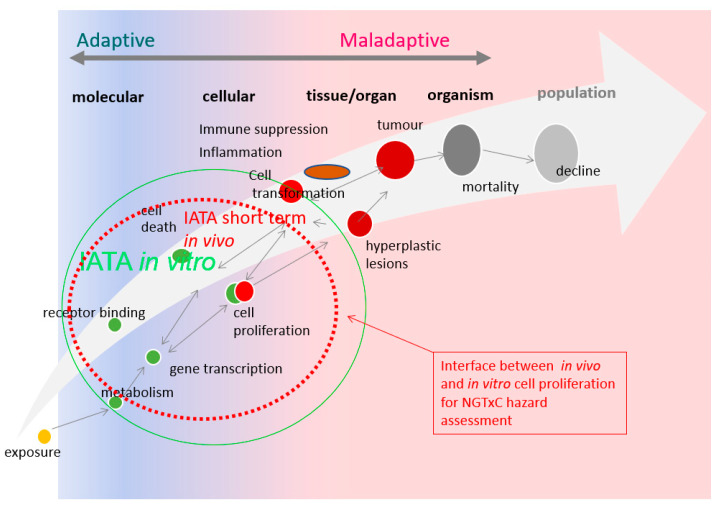
Conceptual overview of the adaptive vs. maladaptive critical data gaps for adverse outcome recognition in NGTxC, and how they can now be overcome using the cell proliferation tools [29]. From adaptive to maladaptive disease progression: key data gaps in the testing and assessment of non-genotoxic carcinogenicity (updated from [29,30]). There are numerous in vitro assays to address the early key events from receptor binding and transactivation, gene transcription, metabolism, and cell proliferation (indicated by the green circle). However, the in vitro cell proliferation assays require further optimization (discussed in Section 2.3), and thus the currently more optimum in vivo cell proliferation tools (discussed in Section 2.2) will need to be utilized in the short-to-medium term and can support the evidence-based shift to in vitro tools in the longer term. A change in morphology represents the point at which adaptive (sustained) proliferation and hyperplasia/dysplasia become maladaptive. Cell proliferation is a key event that is universal to all non-genotoxic pathways known today, and hence an important component of an IATA.

**Figure 2 ijms-24-13246-f002:**
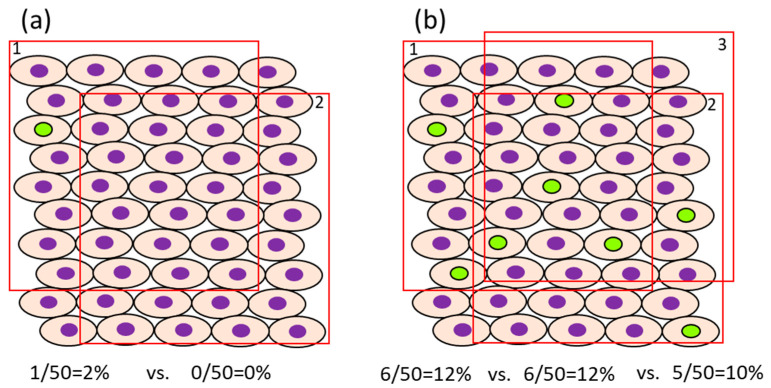
Impact of bias on the evaluation of cell proliferation. The green and purple nucleus indicates BrdU-positive and negative cells, respectively. In case the ratio of BrdU-positive cells is low, the bias caused by the evaluation field would become high (**a**). In case of the high ratio of BrdU-positive cells, the bias caused by the evaluation area would be low (**b**). Original drawing by authors.

**Figure 3 ijms-24-13246-f003:**
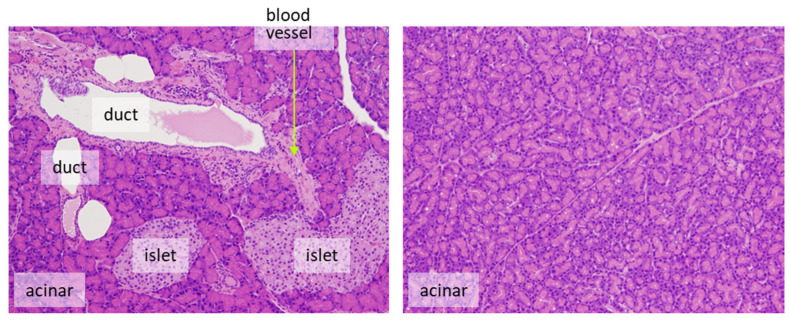
Counting should be performed on target cells. For example, the pancreas is composed of several different components, including acinar cells, islet cells, ductal cells, and other mesenchymal cells, such as blood vessels. Their proportion depends on the field of view. Source: authors. Magnification is the same for both pictures.

**Figure 4 ijms-24-13246-f004:**
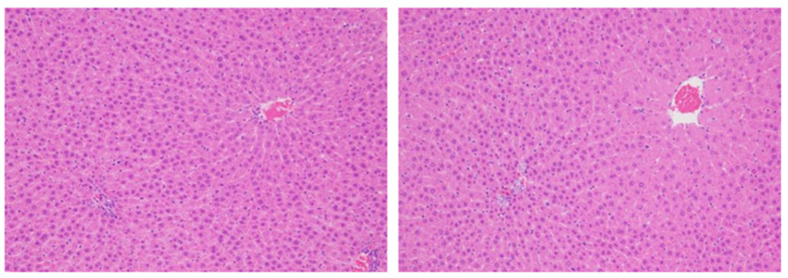
Impact of cell density on evaluation of cell proliferation. In case the proportion and density of cells are uniform, the counting of positive cells might be analyzed per unit area instead of per total number of cells evaluated. However, the size and/or density of the cells may differ, for example, when there is centrilobular hepatocellular hypertrophy (right panel). Source: authors. Magnification is the same for both pictures.

**Figure 5 ijms-24-13246-f005:**
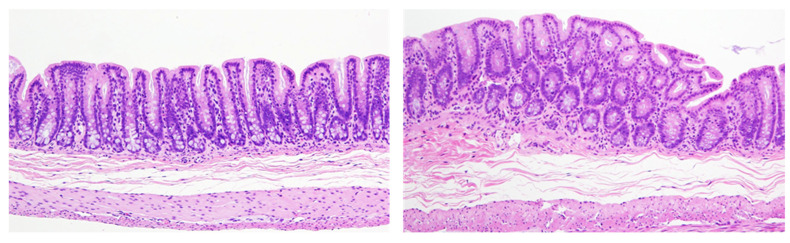
Sections of large intestine show the impact of sectioning on the assessment of cell proliferation. Consideration should be given as to whether the sample trimming and proportion of the cells with proliferative potential is reasonable in the counting area. For example, in the right panel, a non-perpendicular (somewhat angled) slice may occur if the mucosa is slightly wrinkled. Source: authors. Magnification is the same for both pictures.

**Figure 6 ijms-24-13246-f006:**
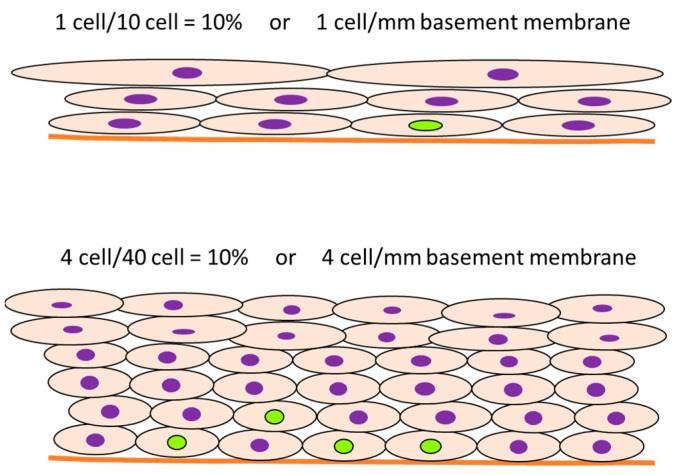
Consideration of positive ratio or number of positive cells per basement membrane or unit area. The green and purple nucleus indicates BrdU-positive and negative cells, respectively. Although the positive ratio is the same in cases 1 and 2, the interpretation from the counting can be different. Original drawing by authors.

**Figure 7 ijms-24-13246-f007:**
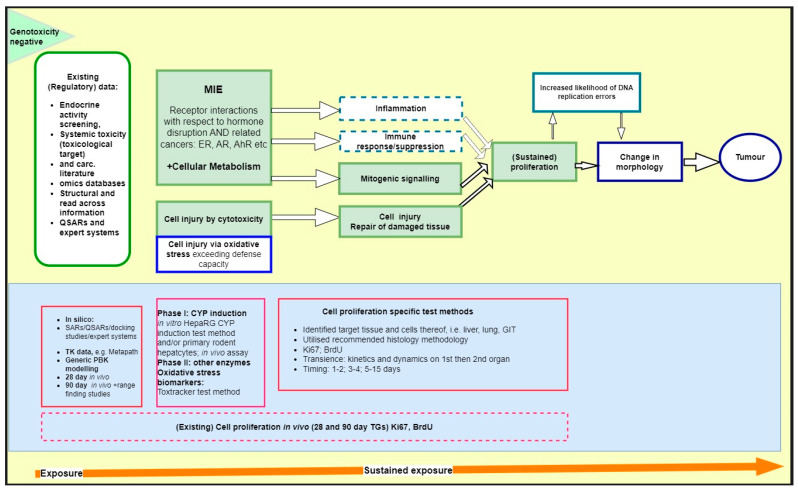
Proposed integration framework for available cell proliferation test methods into the OECD NGTxC IATA (first described in [29]). The focus is on the key events of cell injury and direct mitogenicity leading to (sustained) proliferation (green filled and outlined boxes). Additional key events on immune response and inflammation are indicated in broken blue lines.

**Table 1 ijms-24-13246-t001:** Summary of assay tools/ test methods: readiness for inclusion in the NGTxC IATA.

Ranking	Assay Title	Alias	Critical Aspects and Potential Limitations	Key References
Category A	Proliferation markers (in vivo)	BrdU or Ki-67 in vivo	It is critical that cell number is used as denominator and focused on the correct target cell	[5,32,33,64,65]
Category B/C	In vitro proliferation in primary cells	Serum-free liver mitogen test, CAR assay, BrdU in vitro	Metabolic competence should be characterized; high variability for human tissue; quality of preparation (e.g., presence of non-target cell types); issues regarding primary human cells (e.g., ethics, disease history, contaminants, viruses, number and sex of donors); reproducibility issues	[14,36,66,67,68,69,70,71,72]
	In vitro proliferation in cell lines	DNA synthesis proliferation (BrdU in vitro, ^14^C-thymidine), BRAF inhibitors	Limited metabolic competence and basal proliferation should be taken into account; perturbation of cell signaling due to immortalization	[73,74,75,76,77]
Category C	Assays addressing only cell number (or metabolic activity as a marker proportional to cell number)	CCK8, CellTiter-Glo^TM^, CellTiter 96AQ, CellCiphr^®^ Premier	Cytotoxicity assays, not specific to proliferation	[78,79,80,81,82,83,84,85,86]

**Table 2 ijms-24-13246-t002:** Examples of prototype chemical inducers and non-inducers of cell proliferation in mammals.

Prototypical Activator	Target Organ	Mechanism	Comment	Key References
Phenobarbital	Liver, thyroid	Nuclear receptor binding (CAR), metabolic phase II induction, thyroid (T)-hormone clearance, and constant feedback stimulation	Found in rats and mice	[46,87]
Thiazopyr	Thyroid	Metabolic phase II induction, T-hormone clearance, and constant feedback stimulation	Found in rats	[50]
Clofibrate, Wyeth (WY)-14,643	Liver, pancreas, testis	Nuclear receptor binding (PPARα), metabolic phase II induction, T-hormone clearance, and constant feedback stimulation	Found in rats and mice	[17,48,88]
Chloroform, methapyrilene	Liver, kidney	Cytotoxicity/repair	At cytotoxic doses	[13,44]
Omeprazol, chlorothalonil	Stomach, neuro-endocrine	Gastrin-induced mitogenesis	-	[89]
Folpet, chromium	Duodenum	Cytotoxicity/repair	At cytotoxic doses	[90,91,92]
D-Limonene, nitrapyrin	Kidney	Alpha 2u-globulin	Found in male rats	[93]
Sodium saccharin, ascorbate	Bladder	Crystal formation and chronic local irritation	Found in rats	[94]
Estrogen	Mammary gland	Constant mitogenicity	-	[94]
Cyclosporin A	Lymphoma, squamous cell carcinoma	Immunosuppression	Activation of viral carcinogens, not directly carcinogenic; carcinogenic in Xpa/p53 mice	[95]
Isoniazid	Lung	Mitogen	Found in mice	[96,97]
Fluensulfone, Styrene	Lung	Metabolic induction and resulting damage in Club cells	Found in mice	[52]

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
