# Peer review of "Increased Cell Proliferation as a Key Event in Chemical Carcinogenesis: Application in an Integrated Approach for the Testing and Assessment of Non-Genotoxic Carcinogenesis"

_ijms, 2023, doi:10.3390/ijms241713246_

Round 1

Reviewer 1 Report

The review paper manuscript “Increased cell proliferation as a key event in chemical carcinogenesis: application in an integrated approach for the testing and assessment of non-genotoxic arcinogenesis..

Overall, this is a well written manuscript and has a potential to be accepted.

Nevertheless, the authors should revise better their Abstract, Introduction and Discussion. The study must summarize and clearly present the findings related to previous research in this area.

In addition, several comments follow.

1)General Comment: Please check abbreviations with consistency in main text. Define it at the first appearance, then use it after the definition (e.g. IATA,  NGTxC, CAR, IPCS ,MIE, AIME, etc.)

2) Lines 35-36 Check the font size.

3) Lines 72-76: “There are therefore two ways for an environmental agent to enhance the risk of cancer: …………………..1” Could these both scenarios occur in the same time? If this is the case, you can replace “or” to “or/and”.

4) Please add a new paragraph including further information to introduce these scenarios  e.g. in relation with the tablle2.

5. Please indicate the origin of figures 2,  3,  4, 5, 6, 7.

6. it is proposed to be added a “references” column in the left of the Table 1. If all the information taken from one reference, please highlight it in the table title.

7. Line 567: “be considered (for which OECD guidance is under development)1”. To be updated if needed and added as reference.

8. Please extend and update references (including 2023 findings where needed; e.g regarding omics)

9.Lines 687-693. There is information reported for the first time in Conclusion (e.g see ReCAAP, agrochemical industry, etc.). Please update the manuscript accordingly. Furthermore, it is proposed the Conclusion to be re-written in relation to Abstract and  Introduction taking also acooutn the information presented  in points 2 to 4 of the  current review document.

I will be glad to provide further details if needed and thank you for contacting me.

Moderate editing of English language is considered to be required. 

Author Response

Dear IJMS,  

The authors wish to thank the reviewers for their constructive comments to the manuscript.  

We note that the suggested changes are mostly minor, and that certain comments are already addressed within the manuscript.  

One reviewer has asked for text revisions that represent repetition, whilst the second reviewer hast not made similar suggestions. We have therefore made minor clarifying revisions to the Abstract and Discussion sections. 

As requested, additional references have been included in Table 1. 

We hope that the updates satisfactorily address the reviewer comments.  

Kind regards,  

The authors 

Reviewer 2 Report

This review summarizes current assessment of the non-genotoxic carcinogenesis. This manuscript is well-organized and some points should be revised.

Major points.

#1: In figure 4, is the right image hypertrophy?

#2: In figure 5, is the right image hyperplastic nodule or repaired mucosa after inflammation?

Author Response

(The authors gave the same response as above.)
